# Finite-Frequency Dissipation in Two-Dimensional Superconductors with Disorder at the Nanoscale

**DOI:** 10.3390/nano11081888

**Published:** 2021-07-23

**Authors:** Giulia Venditti, Ilaria Maccari, Marco Grilli, Sergio Caprara

**Affiliations:** Dipartimento di Fisica, Università di Roma Sapienza, Piazzale Aldo Moro, 5, I-00185 Roma, Italy; g.venditti@uniroma1.it (G.V.); ilariam@kth.se (I.M.); marco.grilli@roma1.infn.it (M.G.)

**Keywords:** inhomogeneous superconductivity, nanoscale inhomogeneity, percolation, optical response of superconductors, superfluid stiffness, dissipation in inhomogeneous superconductors

## Abstract

Two-dimensional superconductors with disorder at the nanoscale can host a variety of intriguing phenomena. The superconducting transition is marked by a broad percolative transition with a long tail of the resistivity as function of the temperature. The fragile filamentary superconducting clusters, forming at low temperature, can be strengthened further by proximity effect with the surrounding metallic background, leading to an enhancement of the superfluid stiffness well below the percolative transition. Finite-frequency dissipation effects, e.g., related to the appearance of thermally excited vortices, can also significantly contribute to the resulting physics. Here, we propose a random impedance model to investigate the role of dissipation effects in the formation and strengthening of fragile superconducting clusters, discussing the solution within the effective medium theory.

## 1. Introduction

Two-dimensional (2D) electron systems hosting superconductivity are the object of renewed interest, triggered by new experimental methods and fabrication techniques [1]. Interestingly enough, experiments on 2D systems like SrTiO_3_-based oxide interfaces [2], transition metal dichalcogenides (TMDs) [1], or monolayer graphene [3] have highlighted unusual physical properties that challenge consolidated theoretical schemes. One open intriguing question concerns the possible occurrence of a low-temperature metallic state, which has been claimed to be a new state of matter and dubbed *quantum metal*. This state occurs in systems that are close to a superconducting transition but display a low-temperature saturation or an anomalous behavior of the sheet resistance *R* as a function of the temperature *T* [4,5,6,7,8,9,10,11,12,13,14,15,16,17,18]. Under specific experimental conditions, as those resulting from ionic-liquid gating (in TMDs) or the application of a magnetic field (in graphene), an unusually broad superconducting transition is achieved, whose thermal width cannot possibly be accounted for by standard fluctuational mechanisms [19,20] with physically sound values of the parameters. Another interesting feature is the very slow vanishing of the resistance at low *T*, with a long tail that cannot be understood in terms of standard vortex dissipation. In SrTiO_3_-based oxide interfaces [21,22,23,24] and TMDs [25], these anomalies have been successfully interpreted as the consequence of electron inhomogeneity at the nanoscale and have been described within a 2D random resistor network (RRN) model.

In [25], the main features of the metal-to-superconductor transition in TMDs were captured describing the systems as a 2D superconductor with disorder at the nanoscale, composed by sub-micrometric superconducting puddles with random critical temperature $T_{c}$, embedded in a metallic background, giving a natural explanation to the broadness of the resistive transition. The downturn of the sheet resistance curve R(T) that is observed when the temperature is lowered is understood as being a consequence of the gradual nucleation of superconducting puddles, whose percolation eventually leads to the zero-resistance state. Conversely, if the fraction of these puddles is not large enough to give rise to a percolating superconducting path when T→0, the curve R(T) saturates at low temperatures, and the system stays metallic, thereby providing a simple interpretation for the enigmatic quantum metal. Finally, the long tail of the curve R(T) near the percolative metal-to-superconductor transition is recovered in systems where the puddles are connected by a filamentary superconducting backbone, resulting in a very weak long-distance connectivity of the superconducting cluster.

Once the percolative transition has occurred, the DC transport measurement cannot highlight the properties of the inhomogeneous superconducting cluster at low temperatures. However, the electrodynamic properties of a superconductor with disorder at the nanoscale are expected to be peculiar as well. To investigate the optical conductivity σ(ω) and superfluid stiffness $J_{s}$ of a 2D inhomogeneous superconductor, in a recent work [26] we extended the original RRN to finite frequency, resulting in a random impedance network (RIN). The model was solved within the effective medium theory (EMT) [27,28], and our main result was that if the percolating superconducting cluster is rather fragile and the resulting superfluid stiffness is weak just below the temperature of the percolative transition, it can nonetheless be strengthened (and its superfluid stiffness can be increased) as a result of the proximity effect acting on the nearby metallic background. However, the role and effect of finite-frequency dissipation inside the superconducting cluster was not addressed in [26].

Here, we want to extend our previous investigation, including finite-frequency dissipation effects in the superconducting cluster. Usually, finite-frequency dissipation in the superconducting phase is understood in terms of vortices [29], which can be generated either in the presence of a magnetic field or by thermal excitation. Although very little is known about the nature of thermally generated vortices in a strongly inhomogeneous superconductor with disorder at the nanoscale, it has been shown [30,31] that the presence of spatially correlated disorder may significantly lower the vortex-core energy, inducing an anomalous vortex nucleation below the superconducting transition. More generally, it can be argued that the nucleation of a thermally excited vortex may be particularly cheap if the core of the vortex is located inside a region belonging to the metallic background. As the model proposed in [29], with a proper parametrization, can be re-interpreted as a phenomenological electrodynamic model for finite-frequency dissipation in a superconductor, regardless of the origin of the vortices, we will extend our RIN analysis to include these effects. The conditions of applicability of our description require that the superconducting puddles are large enough to sustain superconductivity, below some local critical temperature. This condition usually makes the capacitance of the superconducting puddles small and the corresponding charging effects negligible, hence we will ignore capacitance effects altogether. As far as the mechanism for finite-frequency dissipation is concerned, we will assume that the superconducting puddles are type II superconductors. Within this respect, we point out that our analysis is carried out in the absence of magnetic field, contrary to the work in [29], so the vortices entailing finite-frequency dissipation are thermally generated within our description. The properties of the RIN will be described within the EMT, to provide a benchmark for more detailed analyses based on the exact solution of the equations of the RIN in the linear response regime. Note that, albeit neglecting space correlations and despite the fact the superconducting puddles are forced to be at least a fraction $\frac{1}{2}$ in order for the system to percolate, the EMT allows for a simple yet insightful description of the temperature dependence of the electrodynamic properties of a superconductor with disorder at the nanoscale.

The paper is organized as follows. In Section 2, we briefly introduce the RRN and RIN models and discuss the EMT approximation. In Section 3, we extend the RIN model to include finite-frequency dissipation in the superconducting cluster by adopting the model proposed in [29] as a phenomenological model. Finally, in Section 4, we summarize our concluding remarks and discuss some possible perspectives.

## 2. A Coarse-Grained Model for Superconductors with Nanoscale Inhomogeneities

A coarse-grained description of a superconductor with disorder at the nanoscale can be achieved in terms of a RRN model [25]. Within this description, the system is imagined as being composed by sub-micrometric regions (puddles) that are large enough to support superconductivity, but may be loosely connected with one another, as they are embedded in a metallic background. Such a metallic matrix can persist down to very low temperatures or become partially superconducting, as a result of the proximity effect, when the temperature is decreased. The system is then mapped onto a 2D square lattice of resistances, each resistor $R_{j}$ representing a metallic region to which a random superconducting critical temperature $T_{c,j}$ is assigned, according to a given probability distribution. When the temperature is lowered, the resistors are gradually switched off, so that the corresponding puddles become superconducting when the condition $T<T_{c,j}$ is met. Therefore, the simplest description of a random resistor is $R_{j}=R_{0}$, when $T>T_{c,j}$, and $R_{j}=0$, when $T\leq T_{c,j}$, $R_{0}$ being the sheet resistance of the system in the metallic state at high temperature ($T>\max\;T_{c,j}$). To model the case where a metallic background persists down to T=0 K, we select a fraction of resistors with $R_{j}=R_{0}$ at all temperatures.

Within the EMT, the sheet resistance of the 2D system $R_{\mathrm{em}}$ is obtained as the solution of the equation [27,28]
(1)$$\sum_{\alpha =s,m}w_{\alpha }\left(T\right)\;\frac{R_{\mathrm{em}}-R_{\alpha }}{R_{\mathrm{em}}+R_{\alpha }}=0,$$
where the subscript s(m) refers to the superconducting (metallic) cluster, $w_{s}\left(T\right)=\int_{T}^{+\infty }dT_{c}\;P_{s}\left(T_{c}\right)$ is the superconducting fraction at a temperature *T*, $P_{s}\left(T_{c}\right)$ is the probability distribution of critical temperatures inside the superconducting cluster (properly normalized to the total superconducting fraction, see below), $w_{m}\left(T\right)=1-w_{s}\left(T\right)$ is the metallic fraction at a temperature *T*, $R_{m}=R_{0}$, and $R_{s}=0$. Equation (Equation 1) can be solved analytically. Let us assume that the total fraction occupied by the superconducting cluster is ${\bar{w}}_{s}\in \left[0,1\right]$, so that the metallic background occupies a fraction ${\bar{w}}_{m}=1-{\bar{w}}_{s}\in \left[0,1\right]$, and that the critical temperature inside the superconducting cluster is distributed according to a Gaussian with mean ${\bar{T}}_{c}$ and variance $\sigma^{2}$,
(2)$$P_{s}\left(T_{c}\right)=\frac{{\bar{w}}_{s}}{\sqrt{2\pi }\;\sigma }\;e^{-\frac{{\left(T_{c}-{\bar{T}}_{c}\right)}^{2}}{2\sigma^{2}}},$$
normalized to the total weight ${\bar{w}}_{s}$ of the superconducting cluster. Then, the solution of Equation (Equation 1) reads
$$\frac{R_{\mathrm{em}}\left(T\right)}{R_{0}}=\theta \left({\bar{w}}_{s}\;\mathrm{erf}\;\frac{T-{\bar{T}}_{c}}{\sqrt{2}\;\sigma }+1-{\bar{w}}_{s}\;\right),$$
where θ(·) is the Heaviside step function and erf(·) is the error function. The variance of the Gaussian distribution (Equation 2) sets the broadening of the percolative transition and controls the downturn of $R_{\mathrm{em}}$. The parameter ${\bar{w}}_{s}$ rules the percolation of the superconducting cluster and, together with σ, controls the presence of a more or less pronounced tail in the percolative transition. Indeed, the percolation threshold for a homogeneous distribution of superconducting puddles is $w_{p}=\frac{1}{2}$, i.e., the percolative transition, if any, occurs at the temperature $T_{p}$ such that $w_{s}\left(T_{p}\right)=\frac{1}{2}$. If the superconducting cluster is not sufficiently dense and ${\bar{w}}_{s}<w_{p}$, the resistance $R_{\mathrm{em}}$ saturates to a finite value when T→0. Therefore, despite neglecting all spatial correlations, the EMT gives a preliminary insight into the main mechanisms at play in an inhomogeneous superconductor, including the occurrence of metallic state at T=0 K, if the superconducting cluster does not percolate. If a percolating superconducting cluster exists, the zero-resistance state is reached below the percolation temperature $T_{p}$.

In the superconducting state, $R_{\mathrm{em}}=0$, so that the DC transport properties are not apt to highlight its inhomogeneous character. To gain insight into the electrodynamic properties of an inhomogeneous superconductor, the RRN model must be extended to finite frequencies ω and the local resistors must be replaced by complex impedances, resulting in a RIN model [26]. The simplest description of a random impedance is $z_{j}=R_{0}+i\omega L_{0}$, when $T>T_{c,j}$, and $z_{j}=i\omega L_{0}$, when $T\leq T_{c,j}$, $R_{0}$ being again the sheet resistance of the system in the metallic state at high temperature, while the inductance $L_{0}$ rules the purely reactive response of the superconducting cluster. For the sake of comparison, in a homogeneous Drude model the complex AC impedance of a metal is $z=R_{0}\left(1+i\omega \bar{\tau }\right)=R_{0}+i\omega L_{0}$, where $R_{0}=m/\left(ne^{2}\bar{\tau }\right)$ is the DC resistance (henceforth, we refer all impedances to unit length and unit cross-sectional area of the conductor), *n* is the carrier density, *e* is the electron charge, *m* is the carrier effective mass, $\bar{\tau }$ is the scattering time, and the inductance is $L_{0}=R_{0}\bar{\tau }={\left(e^{2}J_{s}\right)}^{-1}$ (independent of the scattering time $\bar{\tau }$), where $J_{s}=n/m$ is the superfluid stiffness. The purely reactive response in the superconducting state is achieved setting $R_{0}=0$, i.e., taking the formal limit $\bar{\tau }\to \infty$.

In complete analogy with the case of the RRN in Equation (Equation 1), one can derive the EMT equation for the RIN,
(3)$$\sum_{\alpha =m,s}w_{\alpha }\left(T\right)\;\frac{z_{\mathrm{em}}-z_{\alpha }}{z_{\mathrm{em}}+z_{\alpha }}=0,$$
where $z_{\mathrm{em}}$ is the EMT impedance of the network, while its complex conductance is $g_{\mathrm{em}}=z_{\mathrm{em}}^{-1}=g^{\prime }-ig^{\prime \prime }$ (we drop the subscript em in the real and imaginary parts of $g_{\mathrm{em}}$ to lighten the notation).

## 3. Finite-Frequency Dissipation within an Inhomogeneous Superconductor

The RIN model introduced in the previous section considers a constant inductance $L_{0}$ inside the superconducting cluster [26]. The superfluid stiffness below percolation is $J_{s}\left(T\right)=-\lim_{\omega \to 0}\omega \;g^{\prime \prime }\left(\omega ,T\right)$. As we will fix ω to a constant (low) value, the behavior of $J_{s}\left(T\right)$ can be directly read off the curve $-g^{\prime \prime }\left(\omega ,T\right)$.

To include dissipative effects inside the superconducting cluster, we are led to consider a temperature and/or frequency dependent complex inductance $L_{j}=L_{j}\left(\omega ,T\right)$. In [29], the complex impedance was written as $z=R_{0}+Z_{v}+i\omega L_{0}$, where
$$Z_{v}=A\;\frac{i\omega \tau }{1+i\omega \tau }$$
described the dissipation due to vortices generated by the presence of a magnetic field, *A* is proportional to the strength of the magnetic field (but with the dimensions of a resistance), and τ is the relaxation time for vortices. This expression can be borrowed to represent phenomenologically some finite-frequency dissipation effect in the system even in the absence of an external magnetic field, as we assume throughout this piece of work, considering the dissipation as due to thermally excited vortices. We can recast the expression of the complex impedance assuming a frequency dependent complex inductance $z=R_{0}+i\omega L\left(\omega \right)$, where
$$L\left(\omega \right)=L_{0}+\frac{A\tau }{1+i\omega \tau }.$$

Evidently, for ωτ≫1, the finite frequency dissipation is suppressed. In the superconducting state, where $R_{0}=0$, the reactive part of the response is related to
ReL(ω)=L0+Aτ1+(ωτ)2>L0,
i.e., the reactive response, proportional to the inverse of the inductance, is reduced. However, we point out that in the presence of finite-frequency dissipation, the response of the percolating superconducting cluster is not purely reactive and there is a dissipative response ruled by the term
ReZv=A(ωτ)21+(ωτ)2,
which is vanishingly small for ωτ→0, and tends from below to the constant value *A* for ωτ→∞, thus clarifying the meaning of the parameter *A*, which sets the high-frequency dissipation in the superconducting state.

Of course, within a phenomenological description, the values of the parameters *A* and τ must be adapted to a specific system. As very little is known about thermally excited vortices in strongly inhomogeneous superconductors with a metallic background persisting down to T=0 K, for the sake of illustration, in the following we adopt values of the parameters comparable to those used in [29], taking $A=10^{-3}\;\Omega$ and τ of the order of 1μs. To highlight finite-frequency dissipation we take a frequency ω=0.1 MHz, so that ωτ≈0.1 is small but not negligible. In the following, we discuss the properties of the RIN within the EMT, assuming that the complex impedance is $z_{j}=R_{0}+Z_{v}+i\omega L_{0}$ above the local (random) critical temperature and $z_{j}=Z_{v}+i\omega L_{0}$ below it, while the eventual metallic residue has always $z_{j}=R_{0}+Z_{v}+i\omega L_{0}$.

Let us consider first the simplest case of a system where all the resistors contributes to superconductivity, i.e., ${\bar{w}}_{s}=1$. The parameters of the distribution of critical temperatures (Equation (Equation 2)) are chosen to be ${\bar{T}}_{c}=0.2$ K, σ=0.05 K. Let us fix, for the sake of definiteness, $R_{0}=600\;\Omega$, $L_{0}=1$ nH, as values typical of SrTiO_3_-based oxide interfaces [26], and ω=0.1 MHz, which will allow us to highlight dissipative effects in the forthcoming discussion.

The solution of the EMT equation (Equation (Equation 3)), gives a resistance R=Re(zem) shown as the blue curve in Figure 1b, whose value coincides with the result obtained for the case ω=0 (as $\omega L_{0}\ll R_{0}$). As it can be evinced from the distribution of critical temperatures (red shaded area in Figure 1), despite the fact that the superconducting cluster occupies the whole system, percolation in a 2D system requires only $\frac{1}{2}$ of the puddles to turn superconducting and the percolative transition occurs at $T={\bar{T}}_{c}$. The width of the percolative transition is ruled by σ. No tail is present.

The imaginary part of the complex conductance $-g^{\prime \prime }$ becomes relevant below percolation, where the AC response of the RIN is purely reactive [23] (see Figure 1a). The solution of Equation (Equation 3) with no dissipative effects (A=0) is plotted in yellow. As soon as A>0, finite-frequency dissipation suppresses the reactive response of the system below the percolative transition. In the regime ωτ≪1, once we fixed $A=10^{-3}\;\Omega$, this suppression increases with increasing τ: this is shown in Figure 1a, respectively, for the light blue (τ=0.5μs), green (τ=1μs) and red (τ=5μs) curves. As we are considering a thermal activation mechanism for dissipation, we can expect that *A* depends on *T*. This is shown by the violet curve in Figure 1 labeled with A(T) (τ=5μs), where a dependence $A\left(T\right)=A_{0}\;e^{-({\bar{T}}_{c}-T)/T}$, with $A_{0}=10^{-3}\;\Omega$, is adopted to describe the thermal excitation of vortices, exponentially suppressed at T=0 K. To make the comparison with the case of a constant *A* easier, we assumed that near percolation $A\left(T\right)\approx A_{0}$, so that the curve with variable A(T) collapses onto the curve with $A=A_{0}$, while smoothly joining the curve with A=0 at low temperature.

Let us now consider a more complicated situation in which the system host three components: the metallic background (m), the percolative superconducting cluster (s1), and a smaller superconducting cluster (s2) emerging by proximity effect below the percolative transition. The minimal required fraction of superconducting bonds to have a low-temperature global superconducting state is ${\bar{w}}_{s1}=\frac{1}{2}$. In this case, the resistive transition is characterized by a very pronounced tail because all the puddles must turn superconducting to reach percolation (see the blue curve in Figure 2b). Even after the appearance of a global zero-resistance state, the superconducting cluster is in this case very fragile towards an applied current [32]. Suppose now that, at lower temperatures, a second superconducting cluster is formed, e.g., as a consequence of the proximity effect acting on the metallic background. This second superconducting cluster is much less dense, and occupies a small fraction of the system, say, e.g., ${\bar{w}}_{s2}=\frac{7}{100}$, but it boosts the superfluid stiffness of the fragile superconducting cluster formed at higher temperature. The appearance of this second cluster modifies the reactive response of the system in the superconducting phase.

The EMT equation for this three-component system is
(4)$$\sum_{\alpha =m,s1,s2}w_{\alpha }\left(T\right)\;\frac{z_{\mathrm{em}}-z_{\alpha }}{z_{\mathrm{em}}+z_{\alpha }}=0,$$
where α runs over the three possible states, as in Equations (Equation 1) and (Equation 3): the label m stands for metallic, while s1 and s2 identify the two possible superconducting clusters, whose critical temperatures are distributed according to
$$P_{sk}\left(T_{c}\right)=\frac{{\bar{w}}_{sk}}{\sqrt{2\pi }\;\sigma_{k}}\;e^{-\frac{{\left(T_{c}-{\bar{T}}_{c,k}\right)}^{2}}{2\sigma_{k}^{2}}},$$
normalized to a total weight ${\bar{w}}_{sk}$ for each superconducting cluster sk, with k=1,2.

Following the lines in [26], we assume that, in the absence of the low-temperature superconducting cluster, the inductance of the high-temperature superconducting cluster is ${\bar{L}}_{1}=1$ nH. Assuming that the inductance of the low-temperature superconducting cluster is ${\bar{L}}_{2}=0.5$ nH, we describe the strengthening of the superconducting cluster after proximization with a temperature-dependent inductance of the high-temperature superconducting cluster
$$L_{1}\left(T\right)=\frac{1}{2}\left[{\bar{L}}_{1}\left(1+\mathrm{erf}\;\frac{T-{\bar{T}}_{c,1}}{\sqrt{2}\;\sigma_{1}}\right)+{\bar{L}}_{2}\left(1-\mathrm{erf}\;\frac{T-{\bar{T}}_{c,2}}{\sqrt{2}\;\sigma_{2}}\right)\right],$$
where ${\bar{T}}_{c,1}$ and ${\bar{T}}_{c,2}$ are the average critical temperatures of the high- and low-temperature superconducting cluster, respectively, and $\sigma_{1}^{2},\sigma_{2}^{2}$ are the variances of the corresponding distributions (this description was implemented in [26], but the corresponding formula was inadvertently omitted in the text). We point out that this formula is only meant to interpolate between ${\bar{L}}_{1}$, for $T\gg {\bar{T}}_{c,1}$, and ${\bar{L}}_{2}$, for $T\ll {\bar{T}}_{c,2}$, to phenomenologically describe the strengthening of the superconducting cluster after proximization.

The superconducting fraction of the *k*-th cluster at a temperature *T* is $w_{sk}\left(T\right)=\int_{T}^{+\infty }dT_{c}\;P_{sk}\left(T_{c}\right)$ and $w_{m}\left(T\right)=1-w_{s1}\left(T\right)-w_{s2}\left(T\right)$ is the metallic fraction at a temperature *T*. The corresponding impedances are now $z_{m}=R_{0}+i\omega {\bar{L}}_{1}$, $z_{s1}=i\omega L_{1}\left(T\right)$, and $z_{s2}=i\omega {\bar{L}}_{2}$. Without loss of generality, we assume that ${\bar{T}}_{c,1}>{\bar{T}}_{c,2}$. To describe the enhancement of the stiffness after proximization, we take ${\bar{L}}_{2}<{\bar{L}}_{1}$. Equation (Equation 4) reduces to a third-degree equation in the variable $z_{\mathrm{em}}$, with complex coefficients, whose solution is discussed in detail in [26]. Only one of the three solutions has a positive imaginary part, as it is required for a complex impedance.

The resulting reactive response without dissipative effects is shown in Figure 2, with the yellow curve labeled by A=0. Below the percolative transition, the reactive response of the superconducting cluster stays very small, until the second superconducting cluster nucleates at $T\approx {\bar{T}}_{c,2}$ (the distribution $P_{s2}\left(T_{c}\right)$ is shaded in red), thereby boosting the reactive response of the system. Comparison with Figure 1 shows that now there is a wide temperature gap between the drop of the resistance in the metallic phase above percolation and the rise of the reactive response below percolation.

We start from Equation (Equation 4), where we now take $z_{m}=R_{0}+Z_{v}+i\omega {\bar{L}}_{1}$, $z_{s1}=Z_{v}+i\omega L_{1}\left(T\right)$, and $z_{s2}=Z_{v}+i\omega {\bar{L}}_{2}$. As in the previous case, finite-frequency dissipation suppresses the reactive response in the superconducting state, as one can see in Figure 2 comparing the yellow curve where no dissipation is present (A=0) with the green one, where $A=10^{-3}\;\Omega$ and τ=1μs. If the thermal generation of the dissipative processes is taken into account (see the violet curve labeled by A(T) in Figure 2a), adopting now the expression $A\left(T\right)=A_{0}\;\min\left[e^{-({\bar{T}}_{c,2}-T)/T};1\right]$, with $A_{0}=10^{-3}\;\Omega$, the reactive response interpolates smoothly between the response with no finite-frequency dissipation, at T=0 K, and the response with finite-frequency dissipation, near percolation. We point out that, despite the apparent resemblance with the physics of the vortex matter [33,34], the curves in Figure 2a refer to the case where the magnetic field is absent (as it is assumed throughout our manuscript) and the dissipation in the percolative superconducting state is entirely due to thermally excited vortices.

## 4. Conclusions

Along the line of previous works [25,26,28], we proposed a RIN model to investigate the properties of 2D superconductors with disorder at the nanoscale and discussed the solutions of the model within the EMT. In the present paper, we focused on finite-frequency dissipation effects inside the superconducting cluster, due, e.g., to the thermal excitation of vortices, in the absence of magnetic field. We found that dissipation effects can significantly contribute to reduce and reshape the low-frequency reactive response of the superconducting state that is formed after percolation of the superconducting cluster. We explored the case of a single superconducting cluster, and the case when, once percolation of the first cluster has occurred, a second superconducting cluster is formed, e.g., due to proximization of the metallic background. In the latter case, to keep the analysis as simple as possible, and focus on the main physical mechanisms ruling the interplay between nanoscale disorder and finite-frequency dissipation, we assumed that dissipation is the same in both superconducting clusters. Of course, our RIN model is flexible enough to be equipped with cluster-dependent phenomenological parameters $A_{\alpha }$ and $\tau_{\alpha }$, α=m,s1,s2. Moreover, as soon as experiments will investigate the details of low-frequency dissipation and reactive response in various 2D superconductors with disorder at the nanoscale, like TMDs or SrTiO${}_{3}$-based oxide interfaces, we might be able to refine our analysis and adapt it to a given phenomenological scenario.

We remark that the electrodynamic properties of nanopatterned superconductors, and their exploitation in technological applications and device design, are attracting an ever- increasing interest (for recent examples, see in [35,36]). We think that a better understanding of situations in which the nanopatterning may be random (or self-organized), like the case we are considering, can lead to the manipulation and technological exploitation of superconductors with disorder at the nanoscale.

To conclude, we point out that, despite its simplicity and crudeness, the EMT is apt to highlight the main physical mechanisms observed in superconductors with disorder at the nanoscale, and provide a benchmark for exact solutions of the RIN model obtained, with much more numerical effort, by the solution of Kirchhoff’s and Ohm’s equations in the presence of spatial correlations in the system. Despite the preliminary character of the present work, we have highlighted interesting physical effects associated with finite-frequency dissipation in 2D superconductors with disorder at the nanoscale, that call for further theoretical studies, to include the effect of space correlations, and new experimental investigation of nanopatterned superconductors. 

## Figures and Tables

**Figure 1 nanomaterials-11-01888-f001:**
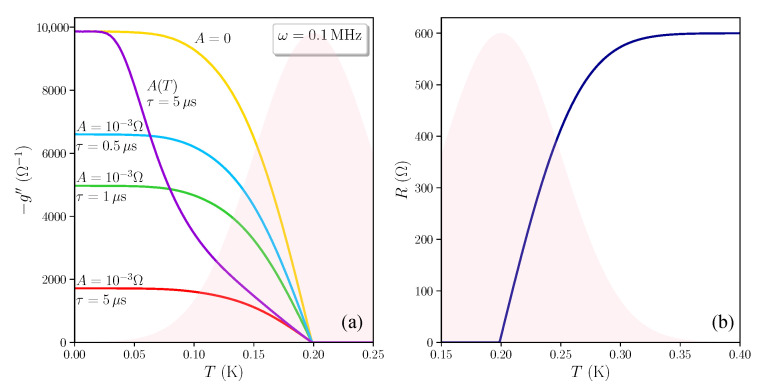
(**a**) Imaginary part of the complex conductance of the RIN within the EMT, $-g^{\prime \prime }$, as a function of the temperature *T*, for $R_{0}=600\;\Omega$, $L_{0}=1$ nH, ω=0.1 MHz. The normal distribution $P_{s}\left(T_{c}\right)$ of the superconducting critical temperatures is shaded in red; the average value and variance are, respectively, ${\bar{T}}_{c}=0.2$ K, σ=0.05 K; the weight is ${\bar{w}}_{s}=1$, i.e., all the resistors participate to superconductivity. The values *A* and τ of the dissipative term $Z_{V}=A\left(i\omega \tau \right)/\left(1+i\omega \tau \right)$ are specified for each curve. In particular, for the violet curve labeled by A(T) we took a temperature dependence $A\left(T\right)=A_{0}\;e^{-({\bar{T}}_{c}-T)/T}$, with $A_{0}=10^{-3}\;\Omega$, to describe the thermal excitation of vortices, which is exponentially suppressed at T=0 K. (**b**) The blue curve gives the sheet resistance of the system above percolation, where the effects of $Z_{V}$ are negligible. All the calculations are carried out assuming zero magnetic field.

**Figure 2 nanomaterials-11-01888-f002:**
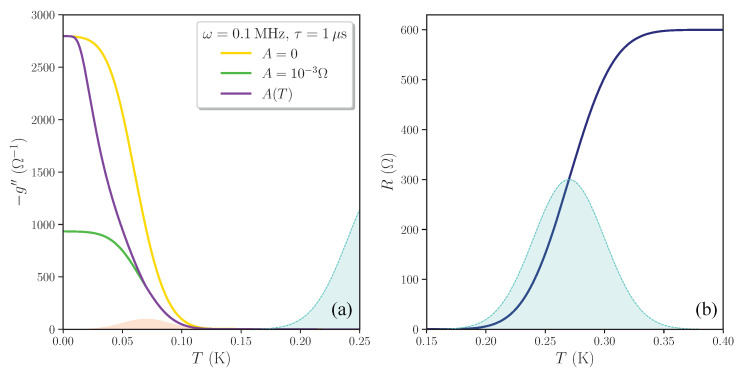
(**a**) Imaginary part of the complex conductance of the three-component RIN within the EMT, $-g^{\prime \prime }$, as a function of the temperature *T*, for $R_{0}=600\;\Omega$, ${\bar{L}}_{1}=1$ nH, ${\bar{L}}_{2}=0.5$ nH, ω=0.1 MHz. The Gaussian distribution of the percolating superconducting cluster $P_{s1}\left(T_{c}\right)$, with mean value ${\bar{T}}_{c,1}=0.27$ K, variance $\sigma_{1}=0.03$ K and weight ${\bar{w}}_{s1}=\frac{1}{2}$, is shaded in light blue; the proximized fraction distribution $P_{s2}\left(T_{c}\right)$ is shaded in red, its parameters are ${\bar{T}}_{c,2}=0.07$ K, $\sigma_{2}=0.02$ K and ${\bar{w}}_{s2}=\frac{7}{100}$. All the three curves are calculated for τ=1μs while the values of *A* are indicated in the legend. For the violet curve we took a temperature dependence $A\left(T\right)=A_{0}\;\min\left[e^{-({\bar{T}}_{c,2}-T)/T};1\right]$, with $A_{0}=10^{-3}\;\Omega$, to describe the thermal excitation of vortices, which is exponentially suppressed at T=0 K. (**b**) The blue curve gives the sheet resistance of the system above percolation, where the effects of $Z_{V}$ are negligible. All the calculations are carried out assuming zero magnetic field.

## Data Availability

The data presented in this study are available on reasonable request from the corresponding author.

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
