# Peer review of "Finite-Frequency Dissipation in Two-Dimensional Superconductors with Disorder at the Nanoscale"

_nanomaterials, 2021, doi:10.3390/nano11081888_

Round 1
Reviewer 1 Report
The authors study a 2d network of random impedances to model an inhomogeneous 2d superconductor. By contrast with previous work, they add a vortex contribution to each impedance phenomenologically. It accounts for the degradation of phase coherence and for dissipation caused by vortex motion. Figure 1 summarizes their results in the case of a metallic type of island and a superconducting type of island, while Figure 2 summarizes their results in the case of a metallic type of island and two different types of superconducting islands. These results are interesting and novel, and they merit publication.
The authors may want to consider the following questions before they resubmit their manuscript, however.
- The phenomenological contribution to the impedance due to vortices, Z_v, was introduced in ref. 29 for 2d superconductors in non-zero magnetic field. The authors "borrow" this phenomenology for the case of thermally excited vortices. Does Figure 1 model the case of zero magnetic field? If so, that should be emphasized throughout the manuscript!
- Does any of the modeling in the paper describe the case of non-zero magnetic field? Figure 2 looks suspiciously close to the phenomenology of 2d vortex matter: a normal metal at high temperature, a Bragg glass at low temperature, and a vortex liquid at intermediate temperature. See, for example, Rodriguez, PRL 87, 207001 (2001) and Rodriguez, PRB 72, 214503 (2005).
- The English can be improved. The use of the phrases "by proximization" and "can interplay" in the abstract are clumsy, for example.
Reviewer 2 Report
Giulia Venditti et al report on the finite frequency dissipation effects associated with the thermally excited vortices. They propose a random impedance model to study the finite frequency dissipation effects in the superconducting clusters. This work is the continuation of the author's earlier work and investigation on this subject where the role and effect of finite-frequency dissipation inside the superconducting cluster were not addressed before. In this work, the authors extended their early investigation and included the finite-frequency dissipation effects in the superconducting cluster. Overall, the paper is well written and is interesting for the condensed matter community. Here are a few minor points for the authors to consider: Page 3, line 1, there seems to be a typo in the first line, Equation 4, where the author talks about ‘where sk, with …’ I can’t see ‘sk’ in the equation. I think both Fig. 1 and 2 could be described in a better way. From the presentation of the figures in the text, it is hard to find what color represents what information and also what color corresponds to what axis (left or right). It would be good if the authors revise these sections clearly addressing each figure (and each color) in the text. It would be useful for the experimental community to know how this investigation would help their device performance. For example, the authors discuss optical properties of 2D high-Tc superconductors but do not suggest any experiment or at least how their theoretical investigation may help improving high Tc superconducting based device performance. What is the application of such studies? The authors may refer to and discuss recent important publication in this subject, for example:
- Proceedings of the IEEE 108 (5), 721-734 (2020).
- PNAS 2021 Vol. 118 No. 4 e2012847118 https://doi.org/10.1073/pnas.2012847118 | 1 of 6
Reviewer 3 Report
I don't think this model is appropriate for describing metal-superconductor response in the presence of disorder in the thermodynamic limit. The main effects ignored are Meissner and skin depth effects. It also does not take into account the nature of the superconducting islands (Ist or IInd type). Modelling such islands as purely inductive elements and ignoring the capacitive component (together with the possibility of the Coulomb blockade) is also too stretchy. Overall, the model completely ignores spatial correlations, and I cannot recommend it for publication.
